Mussel biofiltration effects on attached bacteria and unicellular eukaryotes in fish-rearing seawater

Voudanta Eleni 1
Kormas Konstantinos Ar 2 kkormas@uth.gr kkormas@gmail.com
Monchy Sebastién 1
Delegrange Alice 1
Vincent Dorothée 1
Genitsaris Savvas 1
Christaki Urania 1
1 Laboratoire d’Océanologie et Géosciences (LOG), UMR CNRS 8187, Université du Littoral Côte d’ Opale , Wimereux , France
2 Department of Ichthyology & Aquatic Environment, School of Agricultural Sciences, University of Thessaly , Volos , Greece
Audet Céline
Electronic publication date: 2016 Mar 29
Publication date: 2016
Volume: 4
Electronic Location ID: e1829
Received 2015 Dec 2; Accepted 2016 Feb 28
Copyright: ©2016 Voudanta et al.
Copyright year: 2016
Copyright holder: Voudanta et al.
License: This is an open access article distributed under the terms of the Creative Commons Attribution License, which permits unrestricted use, distribution, reproduction and adaptation in any medium and for any purpose provided that it is properly attributed. For attribution, the original author(s), title, publication source (PeerJ) and either DOI or URL of the article must be cited.
License URL: https://creativecommons.org/licenses/by/4.0/

Keywords: Mussel biofiltration, Fish farming, Sea bass, Bacterial, Unicellular eukaryotes, Biodiversity

Funding: Aquanord S.A. and ULCO/LOG ‘Nord-Pas de Calais’ FRB-DEMO FRB.2013 This work was funded by a collaborative convention between Aquanord S.A. and ULCO/LOG, and the ‘Nord-Pas de Calais’ FRB-DEMO (FRB.2013) project. The funders had no role in study design, data collection and analysis, decision to publish, or preparation of the manuscript.

==============================
Mussel biofiltration is a widely used approach for the mitigation of aquaculture water. In this study, we investigated the effect of mussel biofiltration on the communities of particle-associated bacteria and unicellular eukaryotes in a sea bass aquaculture in southern North Sea. We assessed the planktonic community changes before and after biofiltration based on the diversity of the 16S and 18S rRNA genes by using next generation sequencing technologies. Although there was no overall reduction in the operational taxonomic units (OTU) numbers between the control (no mussels) and the test (with mussels) tanks, a clear reduction in the relative abundance of the top three most dominant OTUs in every sampling time was observed, ranging between 2–28% and 16–82% for Bacteria and Eukarya, respectively. The bacterial community was dominated by OTUs related to phytoplankton blooms and/or high concentrations of detritus. Among the eukaryotes, several fungal and parasitic groups were found. Their relative abundance in most cases was also reduced from the control to the test tanks; a similar decreasing pattern was also observed for both major higher taxa and functional (trophic) groups. Overall, this study showed the effectiveness of mussel biofiltration on the decrease of microbiota abundance and diversity in seawater fueling fish farms.

Introduction

Water column contains a mixture of microscopic particles of various sizes from colloidal non-living material to living microorganisms ranging from viruses, free-living and attached prokaryotes and unicellular eukaryotes (Azam & Malfatti, 2007). Mussels are efficient non-selective filter-feeders. They can filter large volumes of seawater and retain a wide size range of particles (ca. 5–35 µm diameter) such as uneaten feed, phytoplankton and bacteria (Soto & Mena, 1999; Neori et al., 2004). The large biofiltration capacity of suspended mussels has provided rationale for their use in integrated multi-trophic aquaculture (IMTA) systems as an eco-friendly mitigation tool of excess amount of particulate matter (Petersen et al., 2004). In fact, they have been used to control the abundance of pelagic primary producers (Dolmer, 2000), to filter small particles of salmon fish feed and feces (Reid et al., 2010; Irisarri et al., 2015), and to reduce the environmental impact caused by organic wastes in marine fish farming (Lehtinen et al., 1998; Gao et al., 2008) and therefore appear as efficient biomarkers of environmental contamination to assess marine environments quality (Boening, 1999; Brenner et al., 2014). The important role of mussels is also depicted by their use in aquatic polyculture, where salmonid farming is combined with mussel longlines; this method provides an economical and environmental resolution that both reduces organic pollution and enhances shellfish production (Lehtinen et al., 1998; Gao et al., 2008).

Bacteria can be found in the marine environment either as free-living (<10 µm) or particle associated, i.e., attached bacteria (on particles >10 µm) which are often larger and in higher local concentrations than their free-living counterparts (Caron et al., 1982; Acinas, Antón & Rodríguez-Valera, 1999). Attached bacteria contribution to the total bacteria activity is highly variable and depends on both the abundance of attached bacteria and their associated suspended particles concentrations. Apart from bacteria, unicellular eukaryotes constitute the other major biological component of plankton, including a wide array of cell morphologies/sizes and biological traits, which renders this taxonomical group differentially susceptible to mussel filtration.

European sea bass (Dicentrarchus labrax L. 1758, Moronidae, Perciformes) is one of the most cultured marine fish representing a great economic importance for industrial seafood production (FAO, 2014). The health of reared fish is highly susceptible to microbiological risk imposed by bacteria, unicellular eukaryotes including fungi and parasites, as well as viruses that can trigger significant losses in aquaculture production. Additionally, environmental problems induced by coastal eutrophication can have direct (e.g., harmful algal blooms, fish parasites) and indirect (e.g., water hypoxia/anoxia) effects that might reduce the quality of ecosystem services, including aquaculture, fisheries and recreation (Rekker, De Carvlaho Belchior & Royo Gelabert, 2015). Nowadays, among the top ranked scientific questions for social scientists are related to aquaculture effects (Rudd, 2014), such as the disposal of particulate and dissolved wastes resulting from aquaculture installations and activities.

As the microbiological risk for sea bass aquaculture might rise from microorganisms with a wide range of cell morphology, physiology, ecology and biology, and bearing in mind that mussels are not selective feeders in terms of microorganisms but rather on size (e.g., Delegrange et al., 2015), the mitigation effect of mussels biofiltration is not expected to be equally efficient for all the, directly or indirectly, undesired microorganisms in waters associated with aquaculture. In a recent study, Delegrange et al. (2015) evaluated the effect of mussel filtration to dampen the phytoplankton bloom and enhance juvenile sea bass physiological performances. During the same experiment, we focused on structural changes of the particle-associated bacteria and unicellular eukaryotic communities. We specifically evaluated whether mussel biofiltration effectively reduces the overall microbial species richness, also considering the more under-studied eukaryotic communities, as determined by next generation sequencing technologies.

Materials and Methods

Experimental set-up and sampling

The possibility to use mussels as a mitigation tool to prevent phytoplankton spring bloom noxious effects on farmed fish was tested during a 35-day test-case mesocosm experiment in a fish farm in the north of France during the spring phytoplankton bloom (16/04–21/05/2013). The experimental setting has been described in detail in Delegrange et al. (2015) and consisted of three 5 m3 mesocosms. The experimental system was supplied with bulk seawater, i.e., natural coastal North Sea water mixed with seawater released from the Gravelines Power Plant cooling system (France) heated +10 °C compared to in situ temperature and chlorinated (1%). Bulk seawater was filtered by blue mussels (Mytillus edulis) in a biofiltration tank (M) before fueling a test tank (T) containing juvenile sea bass (Dicentrachus labrax). The third tank was used as a control (C), and contained juvenile sea bass reared in bulk seawater (i.e., not biofiltered). In all the tanks, the flow rate was set at 5 m3 h−1 allowing for one turnover every hour, and photoperiod followed natural day:night cycle. Due to logistic constraints of the aquaculture facilities (e.g., number of available tanks, tanks volume, amount of mussels and fish needed and subsequent risk of contamination) a single experiment was carried out. However, from the number of complementary parameters monitored (see next section) and the number of repeated measures over time, observed tendencies and patterns were clearly illustrating mussels’ biofiltration impact on microbiota. Mussel filtration clearance rates based on phytoplankton (abundance and chlorophyll a concentration) were estimated in Delegrange et al. (2015; see Table 2). To compare mussel filtration efficiency and seawater turnover in the T tank, box plot of effective clearance rates based on both chlorophyll a (chl-a) and total phytoplankton abundance were drawn.

Temperature, salinity, pH, dissolved oxygen, turbidity, chl-a were measured twice a week while phytoplankton taxonomy in C and T tanks was investigated at days (d) 0, 3, 10, 17, 24, 31, 35 and are reported elsewhere (Delegrange et al., 2015). In this study, sampling took place at days d0, d7, d14, d21, d28 and d35. Water samples (500–1,000 ml) collected in C and T tanks were screened with a 200 µm mesh to retain larger particles and most metazoans. Samples were then filtered on 3 µm nucleopore filters (47 mm diameter) using a low filtration pressure in order to minimize organism disruption. The filters were immediately frozen in liquid nitrogen and stored at −80 °C until analysis.

Molecular analysis and data processing

Bulk DNA of the retained material on the filters, was extracted and purified, after filtration, with the PowerWater® DNA isolation kit (Mobio Laboratories Inc, Carlsbad, CA, USA), following the manufacturer’s protocol. The quantity of the DNA was between 0.78 and 10.5 ng µL−1 as measured by the Qubit® 2.0 Fluorometer (Thermo Fisher Scientific Inc, Waltham, Massachusetts, USA).

The DNA samples were amplified using the two universal eukaryote primers 18S-82F (5′-ACCAGACTTGCCCTCC-3′) (Lopez-Garcia et al., 2003) and Euk-516r (5′-ACCAGACTTGCCCTCC-3′) (Amann et al., 1990), and two universal bacterial primer S-D-Bact-0341-b-S-17 (CCTACGGNGGCWGCAG) and S-D-Bact-0785-a-A-21 (GACTACHVGGGTATCTAATCC) (Klindworth et al., 2014). The eukaryote primers amplify a domain around 490 bp of the V2–V3 18S rDNA regions, while the prokaryote primers amplify a 465 bp domain from the V3–V4 16S rRDNA regions rDNA. The libraries were constructed by ‘Genes Diffusion’ company (Lille, France). Firstly, Polymerase Chain Reaction (PCR) were carried out, using the two sets of universal primers, according to standard conditions for Platinum Taq High-Fidelity DNA Polymerase (Invitrogen, Carlsbad, CA, USA), with 5 ng of environmental DNA as template, using the GeneAmp PCR System Apparatus (Applied Biosystems, Foster City, CA, USA). The PCR cycle conditions were the following: after the denaturation step at 95 °C for 5 min, 30 cycles of amplification were performed at 95 °C for 30 s, 50 °C for 30 s, and 72 °C for 1 min. A final extension step of 7 min at 72 °C was included. A second PCR was carried out on each sample independently, in order to add a 10 bp (base pairs) specific tag sequence to each sample. Finally, all the amplicons were mixed together stoichiometrically before their sequencing in one run of MiSeq PE 2×300 Illumina (CNRS-UMR8199, Lille).

The sequences were processed using the MOTHUR v1.34.0 software (Schloss et al., 2009) following the standard operating procedure (Schloss, Gevers & Westcott, 2011). Only reads above 492 bp for eukaryotes and 465 bp for bacteria, with homopolymers shorter than 8 bp were kept in the analysis. Eukaryotes and Bacteria sequences were dereplicated to the unique sequences and aligned independently against the SILVA 108 database (Pruesse et al., 2007) containing only the targeted region (V2–V3 for eukaryotes and V3–V4 for bacteria) and matching the universal primers used in this study. Subsequently, around 130 sequences for eukaryotes and 8.956 sequences for bacteria suspected of being chimeras were removed using the UCHIME software (Edgar, 2010). The remaining sequences were clustered into operational taxonomical units (OTUs) at 97% similarity. Single singletons (sequences present only once in one sample) were removed from downstream analysis. Finally, the dataset containing both eukaryote and bacterial sequences was normalized according to the lowest number of reads in a sample (55,102 reads). Rarefaction curves calculated for all the sampling dates approached a plateau in most cases when ≥97% levels of sequence similarities were applied (Fig. S1). Sequencing data from this study have been submitted to the Sequence Read Archive (SRA) (http://www.ncbi.nlm.nih.gov/sra) with accession number SRP061259.

Taxonomic classification was assigned using BLASTN (Altschul et al., 1990), against the sequences (PR2 database containing 23.003 protist sequences (Guillou et al., 2013)) for eukaryotes and against the SILVA database for Bacteria containing 530.946 bacterial sequences (Pruesse et al., 2007). The OTUs identified as metazoans (69 OTUs) were removed from analysis.

For each date, OTU relative abundances between the C and T tanks were compared with the Wilcoxon test. For the whole experiment Chi2 test was used to compare C and T tanks for OTU abundance frequency distributions of eukaryotes taxonomic and trophic groups. Statistical analyses were done using the PAST 3c software (Hammer, Harper & Ryan, 2001).

Results

Environmental parameters

The hydrological parameters were reported in detail in Delegrange et al. (2015). Temperature, salinity and dissolved oxygen, were similar, while turbidity and chl-a were significantly higher in the C-tank (Wilcoxon–Mann–Whitney test, p < 0.001; Delegrange et al., 2015). Chl-a—which is a proxy for phytoplankton biomass—ranged between 0.80 µg L−1 and 10.35 µg L−1 (mean ± sd, 4.97 ± 3.80 µg L−1) in the C-tank, whereas it ranged between 0.19 and 2.28 µg L−1 (mean ± sd, 0.79 ± 0.62 µg L−1) in the T-tank (Table S1 , Delegrange et al., 2015).

Filtration effect

Mussel effective clearance rates based on chl-a concentration removal and phytoplankton abundance decrease reached 11–76 and 26–60 m3 per day, respectively, being at least twice as much as seawater turnover in T tank (Fig. 1). This impacted OTUs relative abundances and changes in the total number of OTUs between C and T tanks were different for Bacteria and Eukarya at the various sampling points (Table 1). The lower bacterial OTU richness in the T tank compared to the C tank, was observed at d0, d14 and d28, while for the Eukarya less OTUs in the T tank were observed at d0, d7, d21, and d35. For the Bacteria, the percentage of shared OTUs (Fig. 2) between C and T tanks ranged only between 8.3% (21 d) and 12.9% (28 d). For the Eukarya this value was twice higher ranging between 15.5% (d28) and 38.7% (d7).

Figure 1 Mussel Effective Clearance Rates (ECR) based on phytoplankton abundance (ECRphyto) decrease and chlorophyll a concentration (ECRchla).

The grey line represents the flow rate in the mussel tank (5∗103Lh−1) and the dashed line stands for Population Maximum Filtration Rate (WCRpop) calculated from Riisgård, Larsen & Pleissner (2014) using mussel mean wet weight (modified from Delegrange et al., 2015).

Although the contribution of rare, common and abundant OTUs (Fig. S2) was similar for both Bacteria and Eukarya at all sampling points, a decrease in the relative abundance of the top three most abundant OTUs was recorded (Fig. 3). The only cases where an increase occurred from the control to the test tank was only for Eukarya on d0 and a much less important increase on d7, for the three most dominant OTUs. In all the rest cases, there was no increase to the extent that an OTU dominated (Fig. 3). The relative abundance of the single most dominant Bacteria OTU varied between 4.6% and 19.3% (Table 1). The dominant Bacteria in the C tank were related to Polaribacter spp. (d7, d14, d28), Sulfitobacter sp. (d0), Pseudoalteromonas sp. (d21) and Arcobacter sp. (d35) (Fig. 3). The relative abundance of the single most abundant Eukarya OTU in the C tank ranged between 15.5% and 96.1% (Table 1) and these were related to Phaeocystis sp. (d0, d7, d21, d28), an unaffiliated Dinophyceae (d14) and Zoothamnium sp. (d35) (Fig. 2). On d7 was the only case where the cumulative relative abundance of the top three most dominant OTU slightly increased from C to T tank (Fig. 3). Regarding Eukarya, mussel biofiltration resulted in decreased OTUs richness in the T tank for all higher level taxa in all trophic groups (Fig. 4) however these higher values did not result to any significant difference of the mean ranks (Wilcoxon test, p > 0.05). However, considering all the sampling dates there were significant differences regarding the frequency distribution of the OTUs abundance between the two tanks, except for Rhizaria and Excavata (Chi2 test, Fig. 4).

Table 1 Qualitative and quantitative normalized richness and abundance of the bacterial (B) and eukaryotic (E) operational taxonomic units (OTUs) in the control (C) and mussels tank (T) at all sampling days (d0-35).

Sampling	Reads	OTUs	Dominance (%) of the most abundant OTUs (most closest relative)	No. of the most dominant OTUs (cumulative relative dominance ≥66.0%)	
		B	E	B	E	B	E	
Cd0	54,147	1,442	410	9.2% (Sulfitobacter sp.)	56.1% (Phaeocystis sp.)	46 (66.1%)	2 (69.3%)	
Td0	53,902	1,345	279	9.7% (Flavobacterialesa)	73.2% (Syndinialesb)	41 (66.6%)	1 (73.2%)	
Cd7	54,950	1,882	189	6.1% (Polaribacter sp.)	94.9% (Phaeocystis sp.)	46 (66.4%)	1 (94.9%)	
Td7	54,892	2,101	180	4.6% (Polaribacter sp.)	96.1% (Phaeocystis sp.)	58 (66.1%)	1 (96.1%)	
Cd14	55,047	2,814	72	4.6% (Polaribacter sp.)	29.4% (Phaeocystis sp.)	81 (66.1%)	3 (79.4%)	
Td14	54,488	1,487	118	8.7% (Photobacterium sp.)	22.9% (Phaeocystis sp.)	25 (66.2%)	8 (66.0%)	
Cd21	54,760	1,310	192	13.0% (Pseudoalteromonas sp.)	15.5% (Phaeocystis sp.)	17 (66.6%)	10 (66.0%)	
Td21	54,994	1,537	63	11.6% (Vibrio splendidus)	33.0% (Syndinialesa)	21 (66.1%)	7 (66.8%)	
Cd28	55,098	2,494	46	8.6% (Polaribacter sp.)	80.5% (Phaeocystis sp.)	77 (66.1%)	1 (80.5%)	
Td28	54,661	2,185	208	7.4% (Pseudoalteromonas sp.)	23.3% (Syndiniales)	57 (66.1%)	6 (67.1%)	
Cd35	51,188	1,129	403	16.0% (Arcobacter sp.)	16.9% (Zoothamnium sp.)	30 (66.0%)	10 (66.2%)	
Td35	54,160	2,452	222	19.3% (Pseudoalteromonas sp.)	16.2% (Colpodella sp.)	64 (66.0%)	10 (66.2%)	
Notes.

a NS9 Marine Group.

b Dinoflagellate Group I Clade 1-X.

Figure 2 Number of shared and unique operational taxonomic units in control (C) and mussels’ tank (T).

Figure 3 Changes in the relative abundance of the three most abundant operational taxonomic units (OTUs) from the control to test tank in every sampling point.

Figure 4 (A) Mean OTU abundance for the 9 super-groups for Control and Test tanks. Chi2 test: ∗p < 0.05, ∗∗p < 0.001 ∗∗∗p < 0.0001, red colour: Fisher test when the conditions of Chi2 were not respected (here when >20% frequencies were 0 or 1); (B) Mean OTU abundance for 6 trophic groups for Control and Test tanks. Chi2 test: ∗∗p < 0.001 ∗∗∗p < 0.0001.

OTUs taxonomic affiliation

In total, 23 phyla/higher taxonomic groups (Fig. S3) included the 15,046 bacterial OTUs were found in both tanks during the experiment. Bacteroidetes alone included 82.1% of total OTUs (12,347), and these OTUs were dominated by members of the Flavobacteriaceae family (73.24% of total OTUs number; 11,021) (Fig. S3). From the rest of the bacterial phyla, the γ- and α-Proteobacteria dominated by including 7.2% (1,084) and 3.9% (593) of the total number of OTUs, respectively (Fig. S2). OTUs were sorted into major trophic groups, such as microplankton grazers, autotrophs, picoplankton grazers, nanoplankton grazers, mixotrophs and parasites.

The 968 eukaryotic OTUs were affiliated into nine ‘super-groups’ and 41 higher ‘taxonomic groups’ distributed in all samples (Fig. S4). Alveolates were the most diverse group accounting for 34.4% of the total OTUs (333 OTUs), followed by Stramenopiles (30.4%, 294 OTUs) and Opisthokonta (17.9%, 173 OTUs). Within, these three supergoups, the most representative ‘taxonomic groups’ in terms of OTU numbers were Ciliata, Labyrithulea, and Fungi. The other six supergroups included from 6 to 52 OTUs. The relative OTU abundance of Stramenopiles was relatively stable in all samples (32.6 ± 3%) . Fungi OTUs were well represented in all samples (10.6 ± 4.2%) except in the C tank at d28 where they were absent. Relative abundance of Alveolata-related OTUs was between 28.6 and 50% and showed highest values in the C tank at d14 and d28 (50 and 41.3%, respectively). The most dominant eukaryotic OTUs (Fig. 3) were related to Phaeocystis sp., Gyrodinium spp., Zoothamnium sp. and some unaffiliated dinoflagellates, bicosidia and ciliates. Finally, a significant positive correlation was observed between the total number of OTUs and the number of parasitic OTUs (Fig. 5; p < 0.01).

Figure 5 Relationship between the number of total eukaryotic and parasitic operational taxonomic units (OTUs) in the control and test tanks in every sampling point.

The determination coefficient R2 and the p value of the regression are also indicated.

Discussion

Mussels filtration efficiency to remove suspended particles including microorganisms (Dame & Dankers, 1988; Asmus & Asmus, 1991) renders the use of these organisms an eco-friendly way to improve water quality in fish farming areas by dampening organic particulate matter concentrations (Burkholder & Shumway, 2011). Delegrange et al. (2015) and Fig. 1 demonstrated the mussel clearance rates to be at least twice the water inflow, resulting in effective phytoplankton bloom dampening although inducing a probable food shortage for mussels. This was congruent with the low mussel condition index (3.43 ±0.45 mg cm−3) measured in the same experiment (Delegrange et al., 2015) and could also be related to the reproductive cycle of the mussels in the North Sea (Riisgård, 2001). In this study, mussel filtration impact was considered regarding changes in species richness and relative abundance of planktonic unicellular eukaryotes and particle-attached bacteria. However, our dataset and experimental design did not allow assessing whether these modifications resulted from partial or total digestion by mussels.

Although, absolute numbers of one species can remain the same and still show a decrease in relative abundance if other species increase (making absolute numbers of organisms more important in the context of infection or toxicological potential), in the present study only the relative abundance of OTUs was taken into consideration. This was because of biases of deep-sequencing, which involve pyrosequencing errors and copy-number variations among taxa (Kunin et al., 2010; Medinger et al., 2010), that can produce non-realistic abundance values (see Genitsaris et al., 2016). We assessed OTUs diversity by using Illumina sequencing of the 18S and 16S rRNA genes on ≥3μm material. The resulting rarefaction curves indicated that for most of the samples, the majority of the existing species richness was revealed (Fig. S1).

Filtration effect

As expected due to the highly dynamic nature of the tanks, i.e., water renewal time and mussel filtration, the bacterial and eukaryotic OTUs showed a great variability with no clear pattern from one sampling to the other (Fig. 2). However, the different effect of mussel filtration was clearly illustrated on bacterial and eukaryotic communities for the richness and the relative abundance of the most dominant OTUs, as these are most likely to persist in the tanks compared to the rest of the OTUs with much lower abundances. The low degree of the bacterial community structure overlap between the control and test tanks (8.3–12.9% of shared OTUs) agrees well with the general notion that ca. 10% of shared bacterial fingerprints occurs between ecologically different or distant habitats (Zinger et al., 2011). This suggests that the two tanks harbor different communities, probably due to mussel filtration. Filtration impact was also shown for higher size spectra (>10 µm), particularly on phytoplankton components (Delegrange et al., 2015). Overall, different bacterial OTUs appeared at each sampling point, possibly due to the fact that various bacterial species occur in particle dominated environments (Simon, Smith & Herfort, 2014) such as the one we investigated. Moreover, there were no major differences in the relative abundance of rare, common and abundant bacterial and eukaryotic OTUs (Fig. S2). This could point to the misleading conclusion that mussel filtration has no important effect on these microorganisms. However, a different view comes up when zooming in to the dominant bacterial OTUs. These dominant OTUs are most likely to represent fundamental species in the fish tanks, or at least represent the most well-adapted (sensu Konopka, 2009) species to the prevailing conditions of the tanks. At all sampling times, when considering the three most abundant OTUs, the decrease for the bacterial and the eukaryotic OTUs was 2–28% and 16–72%, respectively (Fig. 3) implying their important role in the plankton community.

For the eukaryotic community, the mussel filtration effect was different. A much higher and more variable percentage of shared OTUs between control and test tanks (15.5–38.7%) was observed, compared to the attached bacteria. This is probably due to the lower species richness of eukaryotes. Interestingly, while the number of OTUs found in the test tank was in most cases lower than in the control tanks this difference was not significant regarding relative abundance (Wilcoxon test). However, the results of the Chi2 test clearly showed that the frequency distribution of the eukaryotic OTUs resulted from two statistically distinct populations.

A plausible hypothesis is that the differences of the filtration effect can be linked to different traits of the eukaryotic groups, such as the much larger heterogeneity in cell size, morphology, life cycle, metabolic/trophic status (Fenchel, 1988). For this reason, we focused on specific groups, at different taxonomic and trophic/functional levels (Fig. 4).

The frequency distribution of OTUs was significantly different (Chi 2 test, Fig. 4) between the two tanks indicating that each tank can be considered as a different ‘ecosystem’. In a parallel study, Delegrange et al. (2015) showed that mussel filtration significantly reduced the phytoplankton biomass 10 times, shifting towards less deleterious phytoplankton species and reduced water column turbidity 4 times. Here, we elucidated that such changes also take place for other eukaryotes and attached bacteria.

Bacteria

Not surprisingly, many of the taxonomic groups found (Fig. S3) were related to microorganisms attached either on particles or other surfaces. This is enforced by the dominant OTUs at every sampling time in both tanks, as well as by the presence of the overall dominant group, the Flavobacteriaceae (Fig. S3), which are well known to be associated with particles in the marine environment (Simon, Smith & Herfort, 2014). Polaribacter spp. and Sulfitobacter sp. were among the most abundant and frequently occurring OTUs in most of the samples along with Pseudoalteromonas (Table 1 and Fig. 3). The concomitant occurrence of these three species is not surprising since they are among the most important ones during phytoplankton bloom as reported by Choi et al. (2016) in an Antarctic polynya. Recently, a Sulfitobacter species was isolated from the toxic marine diatom Pseudo-nitzschia multiseries (Hong et al., 2015). P. multiseries was not present during the experiment nor during the year-round survey carried out in the fish farm but P. delicatissima and P. pungens were identified (Delegrange et al., 2015). Whether Sulfitobacter holds a similar relationship with these two Pseudo-nitzschia species, as with P. multiseries, remains to be investigated. Moreover, it has also been shown that Sulfitobacter promotes diatom cell division, particularly in coastal environments (Amin et al., 2015), thus enforcing its close association with the diatoms in the tanks. Sulfitobacter has been found to be among the first colonisers of sediment trap collected particles after a phytoplankton bloom (Lecleir et al., 2014) and it is also related to diatom and Phaeocystis bloom termination (as is the case in the tanks we investigated) but even in marine diatom cultures (Schäfer et al., 2002). These clues explain the dominance but also the active association of Sulfitobacter with the marine algal material of the tanks.

Polaribacter is well known to thrive in the North Sea phytoplankton blooms (Xing et al., 2015). It is considered to have a vital role in the decomposition of sulfated polysaccharides (Gómez-Pereira et al., 2012) such as the ones found in phytoplankton cell walls, most likely including Phaeocystis globosa (Murray et al., 2007; Wemheuer et al., 2015) and other algal organic material (Williams et al., 2013; Klindworth et al., 2014). In an Antarctic polynya, Delmont et al. (2014) observed Polaribacter dominance at the end of a Phaeocystis antarctica bloom, just as we did after the P. globosa bloom. The close association of Polaribacter with non-Phaeocystis blooms has also been shown for North Sea diatom blooms (Teeling et al., 2012; Klindworth et al., 2014) and reinforces its dominant role in the mussel tanks. Finally, Polaribacter sp. is one of the major members of the Pseudo-nitzschia associated microbiota and it is known to enhance domoic acid production by Pseudo-nitzschia australis (Sison-Mangus et al., 2014). Thus, the observed abundance reduction of this OTU through the mussel filtration could reduce the relevant risk, as Pseudo-nitzschia has been found to occur in the investigated tanks (Delegrange et al., 2015). The observed reduction of Polaribacter abundance by mussel filtration could be an additional factor to account for in future biofiltration studies.

Pseudolateromonas sp. was also among the dominant bacterial OTUs. This microorganism has been associated with diatoms (Amin, Parker & Armbrust, 2012) but it has been assigned a potential algicidal role (Lee et al., 2000). In our study, it peaked after the diatom bloom (d24), implying its latter role in the tanks.

At the end of the experiment (d35), the dominant bacterium was related to Arcobacter sp. This genus is not frequently occurring in marine plankton as it is usually related to animal pathogens (Vandamme et al., 2005) or faecal material (Maugeri et al., 2004; Ottaviani et al., 2013). The bacterial family that most commonly cause diseases in reared sea bass stocks is represented by Vibrionaceae. The major disease-causing species of this family are Vibrio anguillarum and Photobacterium damselae subsp. piscicida, Vibrio rotiferianus and Vibrio harveyi (Toranzo, Magariños & Romalde, 2005; Austin, 2012). However, other species can affect sea bass such as the Piscirickettsia salmonis, (McCarthy et al., 2005), Aeromonas veronii (Uzun & Ogut, 2015), Pseudoalteromonas spp. (Pujalte et al., 2007). In our data set, some of the above mentioned genera were found (data not shown), however, they were always falling in the rare group (<1% relative abundance).

Eukarya

Plankton can also cause harmful effects to both wild (e.g., Thangaraja, Al-Aisry & Al-Kharusi, 2007; Tang & Gobler, 2009) and farmed fish (Bruno, Dear & Seaton, 1989; Treasurer, Hannah & Cox, 2003). The negative effects of phytoplabkton can be: (1) the production of a non–toxic high biomass (mucus overproduction)—bloom forming, leading to foams or scum, which can deplete oxygen levels inducing fish mortality; (2) the production of potent toxins (e.g., domoic acid by diatoms); and (3) the cause of damage in farmed fish gills, mechanically induced by stress exposure or through the production of haemolytic substances (Sellner, Doucette & Kirkpatrick, 2003). In our study, the unicellular eukaryotic communities were dominated by Phaeocystis sp. The haptophyte P. globosa is the central, recurring species in the phytoplankton bloom in the investigation area (e.g., Stelfox-Widdicombe et al., 2004; Hernández-Fariñas et al., 2014) and was also dominant in the experimental tanks (Delegrange et al., 2015) although it was demonstrated not to impact fish physiological performances (Amara et al., 2013). Among the most abundant eukaryotes, Dinoflagellates relative abundance was reduced by mussel biofiltration (Fig. 3). These organisms are considered major diatom grazers in the plankton community (Grattepanche et al., 2011a; Grattepanche et al., 2011b) and are therefore trophically linked to the diatoms in the tanks.

The ecological role of ciliates lies on their grazing on small sized phytoplankton but also on their free-living vs. symbiotic, commensal or parasitic but never mutualistic life mode (Lynn, 2010). Among the dominant OTUs, the relative abundance of Aspidisca sp. and Euplotes sp. were found to be reduced by mussels filtration. These two surface-associated ciliates are frequent and abundant in marine snow (Artolozaga et al., 1997), estuaries (Mironova, Telesh & Skarlato, 2012), while they also occur as lagoon and wastewater treatment plants epiphytes (Martín-Cereceda, Serrano & Guinea, 1996; Dhib et al., 2013), as well as biofouling microorganisms (Watson et al., 2015). Their high abundance is probably related to the high turbidity levels in the tanks (Delegrange et al., 2015), providing multiple niches to proliferate. The observed turbidity decrease (Delegrange et al., 2015) coincided with the reduction of their relative abundance (Fig. 3).

In this study we observed a positive correlation between OTUs number and parasitic OTUs (Fig. 5). This parasites pool contains endosymbionts of other protists but also parasites of special interest for the aquaculture field, i.e., fungi and ichthyosporeans. However, the survivability of fish parasites in the plankton requires further research. Such parasites could be protozoa, myxosporea, helminths and parasitic crustaceans. They may become causative agents of diseases leading to pathologies and mortality, triggering a decrease of fitness and growth and modifying fish behaviour (Feist & Longshaw, 2008). Several parasites belong to unicellular eukarya. These include Amoebozoa, Apicomplexa, Ciliata, Chlorophyceae, and Euglenophyceae, which have been identified as important pathogens for sea bass. Therefore, their potential as serious skin and gill parasites should be assessed (Alvarez-Pellitero, Sitja-Bobadilla & Franco-Sierra, 1993; Sterud, 2002). The role of such parasites is increasing as new information comes along. For example, some protozoan cysts are now considered to act synergistically with bacterial pathogens (Lambrecht et al., 2015). In our study, on day 35, Zoothamnium sp. was the dominant eukaryotic OTU in the control tank, which, however, was drastically reduced through mussel biofiltration. Although this epibiont ciliate has not been found as a parasite of farmed sea bass it is a well-known parasite of copepods (Burris & Dam, 2014), white shrimp (Vidal-Martínez, Jiménez-Cueto & Simá-Álvarez, 2002) and freshwater carps (Dash, Majumder & Raghu Ramudu, 2015).

Other potential parasites, but not being amongst the dominant OTUs, belonged to the groups of MALV, Labyrinthulomycetes, Pirsonia, Oomyeta, Apicomplexa, Perkinsea, Fungi, and Cercozoa (Fig. S4). Members of the MALV group are enigmatic marine alveolates and are most likely considered as intracellular symbionts or parasites (e.g., reviewed in Skovgaard, 2014). In a recent succession study in the eastern English Channel, Fungi and Cercozoa, were mostly found to co-occur with polysaccharide producing Bacillariophyceae (Christaki et al., 2014). Fungi are possibly related to the polysaccharide degradation of the freshly produced organic material by primary producers (Kimura & Naganuma, 2001). It is known for diatoms that polysaccharides are their main exudates (Myklestad, 1995 and references therein), and these sugars could promote the growth of Fungi. Many Cercozoa are parasites of marine organisms, including large heavily silicified diatoms (e.g., Tillmann, Hesse & Tillmann, 1999), which could explain why Fungi and Cercozoa are detected in bloom situations and are poorly represented in oligotrophic conditions (Georges et al., 2014). Labyrinthulomycetes, which appeared at the end of the experiment, are common osmo-heterotrophic marine protists (López-García et al., 2001) having parasitic, commensalistic, or mutualistic relationships with their hosts. Their presence is congruent with their role in decomposition processes (Collado-Mercado, Radway & Collier, 2010) by colonizing fecal pellets, including under deep-sea conditions (Raghukumar et al., 2004).

Mussel biofiltration is becoming a widely used eco-friendly water purification process in aquaculture. This study is an additional contribution to integrated aquaculture focusing on the effect of mussel biofiltration on individual OTUs of particle-associated bacteria and protists. The decrease of both prokaryotic and eukaryotic OTUs abundance and diversity indicated that mussel biofiltration affects their community structure. The bacterial community was dominated by species closely associated to detritus and/or phytoplankton bloom derived material. The eukaryotic community contained a large number of potential parasites of fish and planktonic microorganisms, whose accurate role in the trophodynamics of planktonic system and their risk on aquaculture requires further research.

Supplemental Information

Supplemental Information 1 Supplementary material

Click here for additional data file.

English Editing (http://www.englisheditor.webs.com) is thanked for its English proofing. We are thankful to the reviewers for their helpful comments that significantly improved the original manuscript.

Additional Information and Declarations

Competing Interests

Author Contributions

Data Availability

The authors declare there are no competing interests.

Eleni Voudanta and Alice Delegrange performed the experiments, analyzed the data, wrote the paper, prepared figures and/or tables.

Konstantinos Ar Kormas conceived and designed the experiments, wrote the paper, reviewed drafts of the paper.

Sebastién Monchy analyzed the data, contributed reagents/materials/analysis tools, wrote the paper, reviewed drafts of the paper.

Dorothée Vincent and Urania Christaki conceived and designed the experiments, contributed reagents/materials/analysis tools, wrote the paper, reviewed drafts of the paper.

Savvas Genitsaris analyzed the data (bioinformatics analysis), wrote the paper, reviewed drafts of the paper.

The following information was supplied regarding data availability:

Sequencing data from this study have been submitted to the Sequence Read Archive (SRA) with accession number SRP061259.

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
