# Peer review of "Mussel biofiltration effects on attached bacteria and unicellular eukaryotes in fish-rearing seawater"

_PeerJ, doi:10.7717/peerj.1829_

## Round 0.1 · original submission · Major Revisions

· Academic Editor

Major Revisions

Comments from Reviewer 1 were quite pertinent and should be taken into account in a revised version.

The authors should also take the following comments into consideration.

Results:
1. Some of the numbers presented in the text do not correspond to those present in Tables of Figures. Line 178 – d35 for bacterial OTUs and absence of mention of d35 for Eukaryote OTUs), line 189 (94.9% should be 96.1%).
2. Figure 4 is never mentioned in the Results. Results should be presented first in the Results section before being discussed (move Lines 265-266 in the Results section and add pertinent information).

The Discussion is sometimes confusing. It is almost never mentioned whether the presentation of dominant groups refers to control or mussels tanks. Dominant groups are described, but the differences between control and mussels tanks are not really discussed. There should be some discussion about the great variability of bacterial or eukaryote groups over time (changes in number of OTUs or in dominance from one day of sampling to another) and how this may affect the interpretation of data.

Edits are needed lines 83 (Delagrange), 143 (Pruesse), 159 (parenthesis to be closed), 170-171 (repetition), 279 (italics missing), 280 “).).”, 347 (are frequent), 479 (Rev Ecol Syst), 505-508 (this reference is not cited in the manuscript), 533-534 (misuse of capital letters), 570 (italics missing), 581 (italics missing), 595-596 (misuse of capital letters), 612 (J Sea), 626 (Oceans Oceanogr), 658-659 (misuse of capital letters),

Reviewer 1 ·

Basic reporting

Mussel biofiltration effects on attached bacteria and unicellular eukaryotes in fish rearing seawater
(#7946)

This is a very interesting study and I applaud the authors for undertaking it. However, before publishing, I would recommend the study needs to be reframed and some additional data presented in order for it to better deliver on its implied end goal, of exploring mitigation potential using biofiltration of particle-associated bacteria and unicellular eukaryote communities to improve the welfare of farmed fish.

The manuscript is generally well written so my focus is mainly on conceptual level science. As a disclaimer, I do not have the background to comment on the sequencing and taxonomic analyses, so I have assumed that this has been done correctly, and leave this aspect to other reviewers.

The study correctly points out in lines 80-84 the mussels are non-selective and mainly filter particles, based on size. So the study question really should be ‘would a reduction in harmful microbiota, be any different than the proportional reduction of particles within the filtering size range’? This is indirectly addressed through the use of relative abundance and some useful data is presented. However, absolute numbers of one species can remain the same and still show a decrease in relative abundance if other species increase. It is the absolute number of organisms that is more important in the context of infection or toxicological potential. Can the absolute values be teased out of the number of reads? Another issue with relative abundance, is that if a species decreases, another must increase. What is increasing in figure 2, and would this affect the fish?

I would recommend that as a minimum, an additional figure be added to the study that shows ether the actual or theoretical particle removal over time based on inflow into the mesocosim, and then discuss the study results in this context. Greater or less than the clearance rate?

While the ecological details in the discussion are interesting, much of it does not add to the objectives of the paper. Of greater benefit would be a discussion on whether harmful micro biota are denatured in the mussel feces. If they are not, this could affect the data presented on apparent removal.

Another concern is that with a tank as one experimental unit, there are no true replicates. Consequently, there should at least be statistical consideration for repeated measures.

Other
Line 107. There would be one turnover every hour not a total renewal. You are likely only getting 2/3 new water in the mesocosm per hour. See: Tvinnerein K. & Skybakmoen S. (1989) Water exchange and self-cleaning in ¢sh rearing tanks. In: Aquaculture: A biotechnology in progress (ed. by N. De Pauw, E. Jaspers, H. Ackefors & N.Wilkens), pp.1041^1047. European aquaculture Society, Brendena, Belgium.

Experimental design

See Basic Reporting section

Validity of the findings

See Basic Reporting section

Reviewer 2 ·

Basic reporting

This is a very interesting, and different, approach to the question of the degree, magnitude and character of bivalve bioremediation. Incorporation of a genomic approach and population shifts adds a considerable dimension to current science

Experimental design

Well described and repeatable. There are no areas that appear insufficiently described.

Validity of the findings

Based on the experimental design and supporting statistics, the experiment has been adequately engineered to correspond to the environmental questions posed. Conclusions are not overstated, and the overall use of a genomic approach vs. a standard pathogen challenge provides a new methodology that is likely to be adopted by others looking at such things as integrated aquaculture setting (multitrophic).

Additional comments

This manuscript is exceptionally well written and is a concise capture of a complex experimental design and genomic investigation.

---

## Round 0.2 · accepted · Accept

· Academic Editor

Accept

The authors took in consideration all comments and questions raised by the reviewers and editor and provided adequate responses. The manuscript can now be accepted for publication.

Reviewer 2 ·

Basic reporting

The revisions have adequately dealt with the suggestions raised during the revision.
The paper is acceptable in its current form.

Experimental design

No comments

Validity of the findings

No comments

Additional comments

Paper will make an interesting and valuable contribution this emerging area of "bioservices".